# Determinants of substance use among adolescents aged 10–19 years in Tanga Region, Eastern Tanzania: A cross-sectional analysis

Sanun Ally Kessy[1,2]*, Innocent B. Mboya[3,4], Shraddha Bajaria[4], Innocent Yusufu[4], Mashavu H. Yussuf[4], Amani Tinkasimile[4], Frank Mapendo[4], Wilbald Anthony[4], Imani Irema[4], Isaac Willey Lyatuu[5], David Sando[5], Azan Nyundo[6], Abbas Ismail[6], John Elyas Mtenga[3], Ndinagwe Lloyd Mwaitete[3], Agustine Marero[7], Abdallah Hussein[7], Jovin R. Tibenderana[3,8], Agness Samwel Mchome[9], Esther Moka[9], Julieth Gaston Bitabo[9], Sakina Magadi Mustafa[9], Ally Ramadhani Kassembo[9], Kisaka Kachua[9], Salome Meshack Materu[9], Maurus Ndunguru[5], Charles Mkombe[9], Stephen Mwandambo[9], Mary Mwanyika Sando[4], Wafaie Fawzi[10]

1 Directorate of Training and Research, Benjamin Mkapa Hospital, Dodoma, Tanzania, 2 Ifakara Health Institute, Dar es Salaam, Morogoro, Tanzania, 3 Department of Epidemiology and Biostatistics, KCMC University, Moshi, Tanzania, 4 Africa Academy for Public Health, Dar es Salaam, Tanzania, 5 Management and Development for Health, Dar es Salaam, Tanzania, 6 The University of Dodoma, College of Health and Allied Sciences, Dodoma, Tanzania, 7 The University of Dodoma, Department of Mathematics and Statistics, Dodoma, Tanzania, 8 Department of Public Health, St. Francis University College of Health and Allied Sciences, Morogoro, Tanzania, 9 Tanga City Local Government Authorities, Tanga, Tanzania, 10 Department of Global Health and Population, Harvard T.H. Chan School of Public Health, Harvard University, Boston, Massachusetts, United States of America

* sahnunally@outlook.com

## Abstract

### Background

Despite adolescents making up a significant part of the global population, they have high rates of substance use, linked to various health problems and risky behaviours and increased risk of negative social, cognitive, and physical consequences. Studies on substance use and related risk behaviours among adolescents in Tanzania are crucial for informing targeted interventions and policies to safeguard the health and development of this vulnerable population. We aimed to assess the prevalence and determinants of substance use among adolescents in the Tanga region, eastern in Tanzania.

### Methods

We conducted a secondary analysis of cross-sectional survey data among 1,262 in-school (n = 1,031) and out-of-school (n = 231) adolescents conducted in Tanga, Tanzania in 2021. Data was collected using the Global School Health Survey (GSHS) questionnaire that included questions on substance use, specifically alcohol, smoking marijuana and tobacco, and recreational drugs like cocaine and heroin. Modified

**Data availability statement:** The data analysed in this manuscript was provided by a third-party source, the Africa Academy for Public Health (AAPH), in Dar es Salaam, Tanzania. The corresponding author do not have a legal permission to share the data used in this study and did not have special access privileges beyond those available to other researchers requesting data directly from AAPH. Interested researchers can request access to these datasets by email to the Chief Executive Officer (CEO) of AAPH at info@aaph.or.tz or writing to the following address: 06 Msonge Street, off Mwai Kibaki Road, 14112 Mikocheni. P.O. Box. 32273, Dar-es-Salaam, Tanzania.

**Funding:** This study was supported by the BOTNAR Foundation. The funding provided by the BOTNAR Foundation made it possible to conduct the study and generate the data used for this analysis. The funders had no role in study design, data collection and analysis, decision to publish, or preparation of the manuscript.

**Competing interests:** The authors have declared that no competing interests exist.

Poisson regression models estimated the factors influencing lifetime and current (within the past 30 days preceding the survey) substance use using Stata software version 17.

## Results

Among 1,262 adolescents, the lifetime and current prevalence of substance use were 8.6% and 3.2%, respectively (4.1% and 0.8% among in-school vs. 29% and 13.9% among out-of-school adolescents). The most reported substance used over a lifetime among both in-school and out-of-school adolescents was smoking (2.3% and 27.7%), followed by alcohol drinking (2% and 10.4%). A higher likelihood of lifetime and current substance use was found among older adolescents (15–19 years), males, and who owned a mobile phone, with a social media account, with history of sexual activity, and out-of-school adolescents.

## Conclusion

Substance use is common among the adolescent population in eastern Tanzania. Policies to keep adolescents in school are instrumental in reducing risk behaviours during adolescence. Targeted behaviour change interventions should be considered for older adolescents (15–19 years), out-of-school, males, and those exposed to sexual activities.

## Background

Although adolescents (individuals 10–19 years of age) constitute one-fifth of the world's population, they are often overlooked in health policies and interventions because they are assumed to be healthy [1,2]. Yet, it is well known that adolescence is the most vulnerable period for developing mental, neurological, and substance use disorders [3]. More attention has been given to HIV/AIDS, and sexual and reproductive health (SRH) services than newly emerging teenage issues such as substance abuse, injuries and accidents, mental health, and road safety. Substance use including alcohol, tobacco, and illicit drugs such as heroin and cocaine, during adolescence is common in sub-Saharan Africa, ranging from 10% to 44% [1]. This is linked with abuse and physical and psychological symptoms such sadness, headaches, altered appetite, weight loss, and sleep disturbances [4,5], and with negative consequences on learning and development [5].

Since the year 2000, there has been a 52% increase in the number of years lost to disability due to mental and substance use problems in Africa [6]. Substance use is a major contributor to the loss of disability-adjusted life years (DALYs) in youth worldwide, with Africa carrying a more than twice higher burden than high-income nations [2,3].

Previous studies on substance use have mainly focused on either in-school or out-of-school adolescents, specifically those aged 15–19 years [7], and have minor consideration on the early adolescent demographic aged 10–14 years, which also

constitutes a group at higher risk of substance use [2–4]. Various factors have been shown to have influence on substance abuse among in-school adolescents. Factors such as adolescents feeling understood by their parents, through empathy and open communication, were protective against regular alcohol or marijuana use, whereas being a male, frequently being bullied by others, and worrying occasionally or almost constantly had a high odds of substance use [8]. This was further supported by research done in the Kilimanjaro region in Tanzania [9], but contrary to studies among out-of-school adolescents that have mainly focused on the age of 15–19 years and excluded younger adolescents [2–4]. Studies on substance use and related risk behaviours among adolescents in Tanzania are scarce, coupled with limited data availability to inform policy and interventions.

Moreover, studies on teenage risk behaviours in various settings focused on a few substances or focus on only one substance (alcohol or smoking tobacco and marijuana) [10,7]. Current data on the burden and determinants of substance use among in- and out-of-school adolescents is critical to inform policy choices and interventions in this demographic. The study aimed to assess the prevalence and determinants of substance use, including alcohol, smoking tobacco and/ or marijuana, and recreational drugs among in- and out-of-school adolescents in the Tanga region, Eastern Tanzania.

## Materials and methods

### Study design and setting

We performed a secondary analysis of cross-sectional survey data of adolescents aged 10–19 years, conducted in 2021 in Tanga City, located in the Tanga region of Northern Tanzania, by the Africa Academy for Public Health (AAPH) in collaboration with the Tanga City Council Health Management Team. The primary aim of the original project was to establish a foundation for consistently collecting adolescent health data across various domains to inform efforts aimed at improving global adolescent health and well-being. Tanga City, the administrative and commercial hub of the region, lies along the Indian Ocean in the northeastern part of Tanzania Mainland. It is situated approximately 354 kilometers north of Dar es Salaam and about 250 kilometers south of Mombasa, Kenya. The city borders Muheza District to the west and south, Mkinga District to the north, and the Indian Ocean to the east. With an estimated population of 305,585 and an annual growth rate of 1.2%, adolescents aged 10–19 years make up about a quarter of the population. The city is ethnically diverse, home to groups such as the Sambaa, Zigua, Bondei, and Digo each comprising less than 20% of the population while over 40% belong to other ethnic backgrounds. The population comprises office workers (17%), shopkeepers (20%), industrial workers (17%), and unemployed individuals (46%). Administratively, Tanga City is divided into 4 divisions, 27 wards, and 181 streets. The city council oversees 108 primary schools (80 government and 28 private) and 44 secondary schools (26 government and 18 private), serving a total of 26,202 students [11].

### Study population

The study population constituted 1,031 in-school and 231 out-of-school adolescents randomly selected from 20 out of 27 administrative wards of Tanga City. In each of the selected wards, one primary and one secondary school were chosen, and students within schools were randomly selected proportional to the school size (based on the number of students), school ownership (private, public), sex (male, female), and age (10–14, 15–19). At the community level, out-of-school adolescents were purposefully selected from the main hotspots in the city, such as public parks, recreational areas, and community centres. Adolescents' peers were also used for the recruitment of out-of-school participants to maximize participation and reduce non-response rates.

### Data collection

Trained research assistants collected the data using the Regional School Health Survey (RSHS) questionnaire, which was adopted and modified from the WHO/CDC Global School Health Survey (GSHS) [12]. The tool has also been widely utilized in various other contexts [13,14]. This tool captures information on various domains of adolescent health such as

sexual behaviour, experiences of violence and abuse, substance use (such as alcohol, tobacco and marijuana smoking, and illicit drugs such as heroin and cocaine), personal cleanliness, diet, and physical exercise. In this study, the GSHS questionnaire was standardized and administered in the local Kiswahili language [15].

## Measures

The dependent variables were the lifetime and current substance use. Lifetime substance use constituted adolescents reporting ever using any substance in their lifetime [16,17]. The substances considered included alcohol, smoking cigarettes, marijuana, or hand-rolled tobacco, and recreational drugs, specifically cocaine and heroin. Current substance use pertained to the use of any substances within the last 30 days preceding the interview [16,17].

The independent variables examined included adolescent socio-demographic and behavioural characteristics. The demographic characteristics were adolescents' age (10–14, 15–19 years), sex (male, female), doing any paid job (yes, no), and school status (in- versus out-of-school). Social and behavioural characteristics included a history of sexual practice, owning a mobile phone, having a social media account, ever discussing sexual matters with parents/guardians, and currently having a sexual partner.

## Statistical analysis

We performed data management and statistical analysis in Stata software version 17. Descriptive statistics summarized data using frequencies and proportions for categorical variables and median with interquartile range (IQR) for continuous variables. The Chi-squared test was used to determine the association between the lifetime and current substance use with participant characteristics. Modified Poisson regression models (generalized linear models with Poisson family and log link function) estimated the relative risk (RR) and 95% confidence interval (CI) for factors associated with lifetime and current substance use. For adjusted analysis, manual stepwise regression was used to identify variables independently associated with substance use. The likelihood ratio test (LR test) was used to evaluate the effect of removal or retention of variables such as age and sex during the multivariable analyses. Additionally, the best fitting model was evaluated using the Akaike Information Criterion (AIC). All statistical tests were considered at a p-value<0.05.

## Inclusivity in global research

Additional information regarding the ethical, cultural, and scientific considerations specific to inclusivity in global research is included in the Supporting Information (**S1 Checklist**).

## Ethical consideration

The study was approved by Tanzania National Institute for Medical Research (NIMR) (NIMR/HQ/R.8a/Vol.IX/3865). Permissions were also sought from the school administration to interview selected in-school adolescents and the city director and selected ward executive officers for out-of-school adolescents. Participants aged ≥18 years provided written informed consent. Younger participants (10–17 years) provided written assent, and parental consent was also obtained. Data collection and storage followed strict protocols and national guidelines to protect participants' rights and confidentiality.

## Results

### Participant characteristics

The median age of 1,262 adolescents (interquartile range, IQR) was 13 (10–14) years. Two-thirds (67.6%) of surveyed adolescents were aged 10–14 and 59% were males. Most adolescents included in this study did not own a social media account (88.6%) with (17%) owning a mobile phone at the time of survey. Few respondents (18.9%) came from female headed households and 81.7% were in-school (**Table 1**).

**Table 1. Background characteristics of in-school and out-of-school adolescents in Tanga, Tanzania (N = 1,262).**

| Characteristics | N (%) |
|---|---|
| **Sex** | |
| Male | 744 (59.0) |
| Female | 518 (41.0) |
| **Age (Years)** | |
| Median (interquartile range) | 13 (10-14) |
| 10-14 | 853 (67.6) |
| 15-19 | 409 (32.4) |
| **Own social media account** | |
| Yes | 144 (11.4) |
| No | 1,118 (88.6) |
| **Done paid job last 12 months** | |
| Yes | 385 (30.5) |
| No | 877 (69.5) |
| **Paid job type (n = 385)** | |
| Farming/Fishing | 58 (15.1) |
| Small business | 247 (64.2) |
| Informal or Casual Labor* | 80 (20.8) |
| **Head of the household** | |
| Mother | 239 (18.9) |
| Father | 680 (53.9) |
| Others# | 343 (27.2) |
| **Own a mobile phone** | |
| Yes | 217 (17.2) |
| No | 1,045 (82.8) |
| **School status** | |
| In-school | 1,031 (81.7) |
| Out-of-school | 231 (18.3) |

*Informal or Casual Labor include house cleaning, cleaning pig pens, fetching water, working as a construction labour, carrying loads, driving donkeys, tutoring, clearing bushes or grass, building houses, collecting and selling scrap metal, washing boats, processing fish, and plumbing.

#Others household heads include staying with an employer, an uncle (younger or elder), a guardian, a stepfather, a brother, or a brother-in-law.

### Self-reported prevalence of substance use

The lifetime and current prevalence of substance use was 8.6% and 3.2%. Among in-school adolescents, the lifetime and current prevalence of substance use was (4.1% and 0.8%) compared to (29% and 13.9%) among out-of-school adolescents. The most reported substance used over a lifetime among both in-school and out-of-school adolescents was smoking (2.3% and 27.7%), followed by alcohol drinking (2% and 10.4%) (**Table 2**).

### Lifetime and current substance use by in-school and out-of-school adolescent characteristics

The prevalence of lifetime and current substance use by participant characteristics is shown in **Table 3**. The lifetime and current substance use prevalence was higher among males (14.1% and 5.4%), older (15–19 years) adolescents

**Table 2. Self-reported prevalence of substance uses among in-school and out-of-school adolescents in Tanga, Tanzania.**

| Substance use | Total (N = 1,262) | In-school (n = 1,031) | Out-of-school (n = 231) |
|---|---|---|---|
| **Ever smoked#, n (%)** | | | |
| Yes | 88 (7.0) | 24 (2.3) | 64 (27.7) |
| No | 1174 (93) | 1007 (97.7) | 167 (72.3) |
| **Ever drank alcohol##, n (%)** | | | |
| Yes | 45 (3.6) | 21(2) | 24 (10.4) |
| No | 1217 (96.4) | 1010 (98.0) | 207 (89.6) |
| **Recreational drugs (cocaine and heroin), n (%)** | | | |
| Yes | 9 (0.7) | 5 (0.5) | 4 (1.7) |
| No | 1253 (99.3) | 1026 (99.5) | 227 (98.3) |
| **Lifetime use*, n (%)** | | | |
| Yes | 109 (8.6) | 42 (4.1) | 67 (29) |
| No | 1153 (91.4) | 989 (95.9) | 164 (71) |
| **Current use**, n (%)** | | | |
| Yes | 40 (3.2) | 8 (0.8) | 32 (13.9) |
| No | 1222 (96.8) | 1023 (99.2) | 199 (86.1) |

#Ever Smoked includes cigarette, marijuana, and hand rolled tobacco in the lifetime.

##Ever drank Alcohol included consuming beer, local brews, and spirits in the lifetime.

*Lifetime substance use consisted of adolescents who reported ever using any substance in their lifetime.

**Current substance use consisted of adolescents who consumed any substance within 30 days preceding the survey.

(22.3% and 9.3%), those who owned a mobile phone (24.4% and 9.7%), engaged in paid job (21.0% and 9.1%), discussed sexual-related issues with their parents (9.9% and 3.4%) and out-of-school (29% and 13.9%), respectively.

## Determinants of lifetime and current substance use

Findings from the adjusted regression analysis show a lower relative risk of lifetime substance use among females (RR: 0.15, 95% CI: 0.05-0.41), those who owned a mobile phone (RR: 0.65, 95% CI: 0.44-0.98), and with access to social media (RR: 0.61, 95% CI: 0.42-0.89). We also found a lower relative risk of current substance use among individuals who discussed sexual issues with parents/guardians (RR: 0.55, 95% CI: 0.30-1.02), and owned a mobile phone (RR: 0.43, 95% CI: 0.23-0.81).

On the contrary, higher relative risk of lifetime substance use was found among adolescents who ever had sex (RR: 2.53, 95% CI: 1.63-3.91), as age increase (RR: 1.25, 95% CI: 1.13-1.38) and those out-of-school (RR: 1.61, 95% CI: 1.05-2.45) and for the current substance use a higher relative risk was found among adolescent adolescents who ever Done payment job last 12 month (RR: 2.21, 95% CI: 0.70-7.03), as age increase (RR: 1.47, 95% CI: 1.22, 1.77) and those out of school (RR: 1.97, 95% CI: 0.84, 4.63). (**Table 4**).

## Discussion

In this study, we report the prevalence and determinants of lifetime and current substance use among in- and out-of-school adolescents in the Tanga region, Eastern Tanzania. Findings show the prevalence of lifetime and current substance use were 8.6% and 3.2%, Among adolescents currently using substances, marijuana (79%), cigarettes (36%), and alcohol (38%) were the most reported substances. A higher relative risk of substance use was among adolescents who reported ever having sexual intercourse, aged 15–19 years, and out-of-school.

**Table 3. Lifetime and current substance use by in-school and out-of-school adolescent characteristics in Tanga, Tanzania.**

| Characteristics | Total, n (%) | Lifetime use*, n (%) | Current use**, n (%) |
|---|---|---|---|
| | (N = 1262) | n = 109 | n = 40 |
| **Sex** | | | |
| Male | 744 (59) | 105 (14.1) | 40 (5.4) |
| Female | 518 (41) | 4 (0.8) | 0 (0) |
| **Age** | | | |
| 10-14 | 853 (67.6) | 18 (2.1) | 2 (0.2) |
| 15-19 | 409 (32.4) | 91 (22.3) | 38 (9.3) |
| **Own a mobile phone** | | | |
| Yes | 217 (17.2) | 53 (24.4) | 21 (9.7) |
| No | 1045 (82.8) | 56 (5.4) | 19 (1.8) |
| **Own a social media account** | | | |
| Yes | 1118 (88.6) | 76 (6.8) | 27 (2.4) |
| No | 144 (11.4) | 33 (22.9) | 13 (9) |
| **Done paid job last 12 month** | | | |
| Yes | 385 (30.5) | 81 (21.0) | 35 (9.1) |
| No | 877 (69.5) | 28 (3.1) | 5 (0.6) |
| **Ever had sexual intercourse** | | | |
| Yes | 123 (9.75) | 55 (44.7) | 33 (26.8) |
| No | 1139 (90.2) | 54 (4.7) | 7 (0.5) |
| **Ever discussed sexual-related issues with parents/guardian** | | | |
| Yes | 800 (63.4) | 79 (9.9) | 27 (3.4) |
| No | 462 (36.6) | 30 (6.4) | 13 (2.6) |
| **Currently have a sexual partner** | | | |
| Yes | 80 (6.3) | 36 (45) | 26 (32.5) |
| No | 1182 (93.7) | 73 (6.2) | 14 (1.2) |
| **School status** | | | |
| In-school | 1,031 (81.7) | 42 (4.1) | 8 (0.8) |
| Out-of-school | 231 (18.3) | 67 (29) | 32 (13.9) |

*Lifetime substance use consisted of adolescents who reported ever using any substance in their lifetime.

**Current substance use consisted of adolescents who consumed any substance within 30 days preceding the survey.

The lifetime (8.6%) and current (3.2%) prevalence of substance use in Tanga region, which align with studies in Dar es salaam, Tanzania [7], and other settings in SSA [18]. Smoking and alcohol use were consistently the most common substances used by the adolescent population [19]. In SSA, a much higher lifetime and current substance use prevalence was reported, ranging from 16.4% in Ethiopia to 87.3% in Nigeria [10,20–26]. This difference may be due to variation in cultural norms and attitudes towards substance abuse across countries and strong social support systems, differences in cultural perceptions of substance use across various communities may contribute to variations in prevalence rates and patterns of use. Future research should explore these cultural dimensions in greater depth, including how societal attitudes, family expectations, and peer influences shape adolescent substance use behaviors. Furthermore, countries such as Ethiopia, where substances like Khat are legal, is likely to have a much higher prevalence (>40%) [10] than other countries, where such substances are illegal. The acceptance or normalization of substance use within a society, reflected in prevailing attitudes, can also contribute to a higher prevalence of substance use as adolescents may be less deterred by social disapproval or legal consequences. This highlights the importance of tailoring substance abuse prevention and

intervention strategies to the specific socio-cultural contexts of a particular region. In addition, understanding the impact of existing policies and industry practices related to alcohol and tobacco use can inform strategies to mitigate adolescent substance use, fostering a more effective public health response in the specific country context.

Female adolescents exhibited notably lower prevalence of substance use than males. As documented elsewhere in SSA, this may be contributed by protective parental oversight, greater risk perception among females and adherence to societal expectations for female behaviour [18]. Furthermore, as age increase displayed higher lifetime and current substance use than their younger counterparts. This is likely attributed to greater independence, coping mechanisms for transitional challenges and those dealing with distress and limited parental supervision [7]. These findings are in sync with studies in SSA [7,25,27] and suggest the need for targeted interventions addressing the specific challenges faced by older adolescents. It enforces the importance of parental involvement, robust support systems and age and sex appropriate strategies to mitigate substance use risks during this transitional phase to adulthood.

Adolescents who reported having sexual experiences in the 12 months prior to the survey showed higher rates of both lifetime and current substance use. This can be linked to common underlying risk factors such as a tendency toward risk-taking behaviour's, sensation-seeking, and a lack of strong protective factors like family support and positive role

**Table 4. Crude and Adjusted analysis for Determinants associated with lifetime and current substance use in Tanga, Tanzania (N = 1,262).**

| Characteristics | Lifetime substance use[a] | | Current substance use[b] | |
|---|---|---|---|---|
| | CRR (95%CI) | ARR (95%CI) | CRR (95%CI) | ARR (95%CI) |
| **Sex[#]** | | | | |
| Male | 1.00 | 1.00 | | |
| Female | 0.05 (0.02, 0.14) | 0.15 (0.05, 0.41) | N/A | N/A |
| **Age*** | 1.51(1.40, 1.63) | 1.25 (1.13, 1.38) | 1.90 (1.69, 2.14) | 1.47 (1.22, 1.77) |
| **Own mobile phone** | | | | |
| No | 1.00 | 1.00 | 1.00 | 1.00 |
| Yes | 4.55 (3.22, 6.43) | 0.65 (0.44, 0.98) | 5.31 (2.90,9.712) | 0.43 (0.23, 0.81) |
| **Own social media account** | | | | |
| No | 1.00 | 1.00 | 1.00 | 1.00 |
| Yes | 0.29 (0.20, 0.43) | 0.61 (0.42, 0.89) | 0.27 (0.14, 0.51) | 0.59 (0.32, 1.10) |
| **Done payment job last 12 month[##]** | | | | |
| No | 1.00 | 1.00 | 1.00 | 1.00 |
| Yes | 6.59 (4.36, 9.96) | 1.78 (1.07, 2.97) | 20.50 (7.34, 57.22) | 2.21 (0.70, 7.03) |
| **Ever had sexual intercourse** | | | | |
| No | 1.00 | 1.00 | 1.00 | 1.00 |
| Yes | 9.43 (6.81,13.07) | 2.53 (1.63, 3.91) | 43.65 (19.72, 96.61) | 5.48 (1.89, 15.86) |
| **Ever discussed sexual issues with parents/guardian** | | | | |
| No | 1.00 | 1.00 | 1.00 | 1.00 |
| Yes | 1.52 (1.01, 2.28) | 0.81 (0.57,1.14) | 1.34 (0.69, 2.62) | 0.55 (0.30, 1.02) |
| **School status** | | | | |
| In-school | 1.00 | 1.00 | 1.00 | 1.00 |
| Out-of-school | 7.12 (4.97, 10.19) | 1.61 (1.05, 2.45) | 17.85 (8.33,38.24) | 1.97 (0.84, 4.63) |

[#]*Abbreviations: CRR, Crude Relative Risk; ARR, Adjusted Relative Risk. Estimates for current substance use for sex was not included because there was no female reported to use.*

[##]*The reduction in the coefficient from 20 to 2 is likely due to the inclusion of the variable "Ever had sexual intercourse" which may act as a confounder, influencing the association between substance use and other factors.*

[a]*Lifetime substance use consisted of adolescents who reported ever using any substance in their lifetime.*

[b]*Current substance use consisted of adolescents who consumed any substance within 30 days preceding the survey.*

models. These factors make adolescents more vulnerable to engaging in risky behaviours, including substance use. This is in accordance with the findings from studies in Northern Tanzania [28] among youth aged 15–24 years who reported using alcohol and having a higher likelihood of engaging in risky sexual behaviours [29]. These findings emphasize the need for integrated interventions that address both risky sexual behaviours and substance use among adolescents. Understanding these shared risk factors allows for the development of comprehensive prevention strategies that target multiple interconnected behaviours, promoting overall adolescent well-being.

Out-of-school adolescents in our study had higher odds of lifetime substance use than those currently in school. Unlike out-of-school adolescents, those in school often have structured schedules and are under the supervision of teachers and staff, hence less likely to engage in high-risk behaviours such as experimenting with substances. A higher likelihood of substance uses among out-of-school adolescents, and especially those who engaged in paid jobs, has also been reported in Malawi, where young men who dropped out of school due to poverty, and engaged in paid work were able to afford paying for substances [30]. School nonattendance has also been highlighted as one of the causes of harmful adolescent substance use [10]. Policy initiatives focusing on keeping adolescents in school, and vocational training and economic empowerment programs for vulnerable populations can contribute to reducing the risk of unwanted behaviours and practices, including substance use.

## Study strengths and limitations

The study involved a sample of in-school and out-of-school adolescents aged 10–19 years, which makes it possible to generalize the findings to adolescents from similar settings. Findings on the burden and determinants from this study provide a crucial baseline data to inform interventions and policy decisions among adolescents in Tanzania and other similar settings. Notwithstanding these strengths the study had some drawbacks. Although the out-of-school sample was recruited through community-based methods, which may carry some risk of selection bias, efforts were made to reach diverse participants. This potential limitation should be considered when interpreting differences in substance use, though the findings remain valuable for highlighting disparities between groups. Substance use was self-reported, posing the risk of recall and social desirability bias as well as underreporting of substance use. To mitigate these biases, we ensured anonymity, employed validated tools, and trained interviewers for a supportive environment and enhancing the accuracy of our data. Also, the study could not establish a temporal relationship between the exposures and substance use further studies (longitudinal studies) are required for better understand the causal pathways. While past research has linked substance use to behavioural and psychological factors [9,10,31], these aspects are not typically covered in the GSHS surveys, hence could not be analysed in this study. Finally the study was conducted in Tanga City, which may not be fully representative of other regions in Tanzania or sub-Saharan Africa.

## Conclusion and recommendations

Substance use is common in this adolescent population in Eastern Tanzania. Policies to keep adolescents in school are instrumental in reducing risk behaviours during adolescence. Targeted behaviour change interventions should be given to older adolescents (15–19 years). Possible approaches to achieve this may involve offering educational assistance, providing vocational training for those who are not in school, and enhancing school engagement through mentoring and counselling initiatives. In addition, targeted behaviour modification programs should focus on high-risk groups such as older adolescents (ages 15–19), youth who are not enrolled in school, males, and individuals who have participated in sexual activities or exhibit other risk factors. Also, the study recommends utilizing digital interventions which provide a valuable means for encouraging healthy habits and informing teenagers about the dangers of substance use. Social media platforms can be utilized to spread educational information, promote positive messages, and increase awareness regarding the impacts of substance use, thereby ensuring extensive reach and involvement with this demographic.

## Supporting information

**S1 Checklist. Inclusivity in global research.**
(DOCX)

## Acknowledgments

The authors extend their sincere gratitude to the study participants whose invaluable participation greatly contributed to this research. Special thanks are also extended to the dedicated schoolteachers, the Tanga City Council, project implementers.

## Author contributions

**Conceptualization:** Innocent B. Mboya, Innocent Yusufu.

**Formal analysis:** Sanun Ally Kessy, Innocent B. Mboya.

**Methodology:** Sanun Ally Kessy, Innocent B. Mboya, Shraddha Bajaria, Innocent Yusufu, Mashavu H. Yussuf.

**Supervision:** Innocent B. Mboya, Innocent Yusufu, Mashavu H. Yussuf, Amani Tinkasimile, Frank Mapendo, Mary Mwanyika Sando, Wafaie Fawzi.

**Writing – original draft:** Sanun Ally Kessy.

**Writing – review & editing:** Sanun Ally Kessy, Innocent B. Mboya, Shraddha Bajaria, Innocent Yusufu, Mashavu H. Yussuf, Amani Tinkasimile, Frank Mapendo, Wilbald Anthony, Imani Irema, Isaac Willey Lyatuu, David Sando, Azan Nyundo, Abbas Ismail, John Elyas Mtenga, Ndinagwe Lloyd Mwaitete, Agustine Malero, Abdallah Hussein, Jovin R. Tibenderana, Agness Samwel Mchome, Esther Moka, Julieth Gaston Bitabo, Sakina Magadi Mustafa, Ally Ramadhani Kassembo, Kisaka Kachua, Salome Meshack Materu, Maurus Ndunguru, Charles Mkombe, Stephen Mwandambo, Mary Mwanyika Sando, Wafaie Fawzi.

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
