## [Decision Letter · Decision Letter 0]

PGPH-D-24-02968

Determinants of Substance Use Among Adolescents Aged 10-19 Years in Tanga Region, Eastern Tanzania: A Cross-Sectional Analysis.

Dear Dr. Kessy,

Thank you for submitting your manuscript to PLOS Global Public Health. After careful consideration, we feel that it has merit but does not fully meet PLOS Global Public Health’s publication criteria as it currently stands. Therefore, we invite you to submit a revised version of the manuscript that addresses the points raised during the review process.

Please note that we have only been able to secure a single reviewer to assess your manuscript. We are issuing a decision on your manuscript at this point to prevent further delays in the evaluation of your manuscript. Please be aware that the editor who handles your revised manuscript might find it necessary to invite additional reviewers to assess this work once the revised manuscript is submitted. However, we will aim to proceed on the basis of this single review if possible. 

We look forward to receiving your revised manuscript.

Kind regards,

Annesha Sil, Ph.D.

Staff Editor

Journal Requirements:

1.Please include a complete copy of PLOS’ questionnaire on inclusivity in global research in your revised manuscript. Our policy for research in this area aims to improve transparency in the reporting of research performed outside of researchers’ own country or community. The policy applies to researchers who have travelled to a different country to conduct research, research with Indigenous populations or their lands, and research on cultural artefacts. The questionnaire can also be requested at the journal’s discretion for any other submissions, even if these conditions are not met. Please find more information on the policy and a link to download a blank copy of the questionnaire here: https://journals.plos.org/globalpublichealth/s/best-practices-in-research-reporting. Please upload a completed version of your questionnaire as Supporting Information when you resubmit your manuscript.

2. In the online submission form, you indicated that “The data of this study will be given out upon reasonable request to the other”.

a. In a public repository,

b. Within the manuscript itself, or

c. Uploaded as supplementary information.

Additional Editor Comments (if provided):

Reviewers' comments:

Reviewer's Responses to Questions

**Comments to the Author**

1. Does this manuscript meet PLOS Global Public Health’s publication criteria ? Is the manuscript technically sound, and do the data support the conclusions? The manuscript must describe methodologically and ethically rigorous research with conclusions that are appropriately drawn based on the data presented.

Reviewer #1: Partly

2. Has the statistical analysis been performed appropriately and rigorously?

Reviewer #1: Yes

3. Have the authors made all data underlying the findings in their manuscript fully available (please refer to the Data Availability Statement at the start of the manuscript PDF file)?

Reviewer #1: Yes

4. Is the manuscript presented in an intelligible fashion and written in standard English?

Reviewer #1: Yes

5. Review Comments to the Author

Reviewer #1: Peer Review Report

Manuscript Title: Determinants of Substance Use Among Adolescents Aged 10-19 Years in Tanga Region, Eastern Tanzania: A Cross-Sectional Analysis.

The manuscript presents a cross-sectional analysis of substance use among adolescents in the Tanga region of Tanzania. The study addresses an important public health issue, particularly in a low-resource setting where substance use among adolescents is a growing concern. The authors provide valuable insights into the prevalence and determinants of substance use, with a focus on both in-school and out-of-school adolescents. The study is well-structured, and the methodology is generally sound. However, there are several areas where the manuscript could be improved to enhance its clarity, rigor, and impact.

Major Weaknesses and Areas for Improvement:

1. The study includes 1,262 adolescents, with 1,031 in-school and 231 out-of-school participants. While the sample size is adequate, the relatively small number of out-of-school adolescents (18.3% of the total sample) may limit the generalizability of the findings to this subgroup. The authors should discuss this limitation and consider whether the sample size is sufficient to draw robust conclusions about out-of-school adolescents.

The study was conducted in Tanga City, which may not be representative of other regions in Tanzania or sub-Saharan Africa. The authors should acknowledge this limitation and suggest caution in generalizing the findings to other settings.

2.The study relies on self-reported data, which is prone to recall bias and social desirability bias. While the authors mention efforts to mitigate these biases (e.g., ensuring anonymity, using trained interviewers), they should discuss the potential impact of these biases on the results. For example, underreporting of substance use, particularly among in-school adolescents, could affect the accuracy of the prevalence estimates.

3. The cross-sectional design of the study limits the ability to establish causal relationships between the determinants and substance use. The authors should explicitly state this limitation and suggest that longitudinal studies are needed to better understand the causal pathways.

4. The study does not explore psychological or behavioral factors (e.g., mental health, peer influence, family dynamics) that are known to influence substance use.

5. The authors use modified Poisson regression models to estimate the relative risk (RR) of substance use. While this is appropriate for binary outcomes, the interpretation of the RRs, particularly for current substance use, is challenging due to the small sample size in some subgroups (e.g., only one female reported current substance use). The authors should discuss the potential instability of these estimates and consider alternative statistical approaches, such as logistic regression, which may provide more stable odds ratios (ORs).

6. The study briefly mentions cultural norms and attitudes toward substance use but does not explore these factors in depth. Given the importance of cultural context in shaping substance use behaviors, the authors should consider incorporating a more detailed discussion of how cultural factors may influence the findings.

7. While the study provides valuable policy recommendations, such as keeping adolescents in school and targeting high-risk groups, the authors could expand on how these interventions might be implemented in practice. For example, what specific strategies could be used to keep adolescents in school? How can digital interventions be effectively leveraged to promote healthy behaviors?

Minor Issues

1. The definitions of "lifetime" and "current" substance use are clear, but the authors should ensure that these terms are consistently used throughout the manuscript. For example, in Table 2, the terms "ever smoked" and "ever drank alcohol" are used, which could be confusing. It would be helpful to align these terms with the definitions provided in the text.

2. The results section is well-organized, but some tables could be more reader-friendly. For example, Table 3 could be simplified by removing redundant information and focusing on the key findings. Additionally, the authors should consider using visual aids (e.g., bar charts or graphs) to illustrate the prevalence of substance use across different subgroups.

3. The discussion section is thorough, but it could benefit from a more critical analysis of the findings. For example, the authors should discuss why older adolescents (15-19 years) are more likely to use substances and how this relates to their greater independence and exposure to risk factors.

4. The conclusion is concise but could be expanded to include a broader discussion of the implications for public health policy and practice. For example, how can the findings be used to inform national or regional strategies to reduce substance use among adolescents?

6. PLOS authors have the option to publish the peer review history of their article (what does this mean? ). If published, this will include your full peer review and any attached files.

**Do you want your identity to be public for this peer review?** For information about this choice, including consent withdrawal, please see our Privacy Policy .

Reviewer #1: No

---

## [Decision Letter · Decision Letter 1]

PGPH-D-24-02968R1

Determinants of Substance Use Among Adolescents Aged 10-19 Years in Tanga Region, Eastern Tanzania: A Cross-Sectional Analysis.

Dear Dr. Kessy,

Thank you for submitting your manuscript to PLOS Global Public Health. After careful consideration, we feel that it has merit but does not fully meet PLOS Global Public Health’s publication criteria as it currently stands. Therefore, we invite you to submit a revised version of the manuscript that addresses the points raised during the review process.

A second reviewer has assess the revised manuscript and provided some additional comments to help improve the manuscript. Please review their comments below and make the appropriate revisions to address their concerns.

We look forward to receiving your revised manuscript.

Kind regards,

Emma Campbell, Ph.D

Staff Editor

Journal Requirements:

Reviewers' comments:

Reviewer's Responses to Questions

**Comments to the Author**

1. If the authors have adequately addressed your comments raised in a previous round of review and you feel that this manuscript is now acceptable for publication, you may indicate that here to bypass the “Comments to the Author” section, enter your conflict of interest statement in the “Confidential to Editor” section, and submit your "Accept" recommendation.

Reviewer #2: (No Response)

2. Does this manuscript meet PLOS Global Public Health’s publication criteria ? Is the manuscript technically sound, and do the data support the conclusions? The manuscript must describe methodologically and ethically rigorous research with conclusions that are appropriately drawn based on the data presented.

Reviewer #2: (No Response)

3. Has the statistical analysis been performed appropriately and rigorously?

Reviewer #2: (No Response)

4. Have the authors made all data underlying the findings in their manuscript fully available (please refer to the Data Availability Statement at the start of the manuscript PDF file)?

Reviewer #2: (No Response)

5. Is the manuscript presented in an intelligible fashion and written in standard English?

Reviewer #2: (No Response)

6. Review Comments to the Author

Reviewer #2: I did not review the original manuscript and will only make a few minor comments. The authors have adequately responded to the points raised in the prior reviews. This manuscript presents finding from data collected on adolescents in the Kanga region of Tanzania. It is nice to see this kind of data being collected and analyzed, and it will, I hope, be part of an ongoing and larger effort to monitor and respond to health risk behaviors in Tanzania and other similar sub-Saharan African countries. The authors made an effort to recruit a sample representative of the in-school population of students in the Kanga region. The recruitment methods for the out-of-school sample are perhaps more vulnerable to selection bias, which is a limitation that should perhaps be highlighted more clearly, since a primary finding of the study is the difference in substance use between the in-school and out-of-school groups. Below are few other points for the authors to consider:

1. It would helpful for readers to be given some brief information on the Kanga province and how it is similar to or different from other parts of Tanzania and other parts sub-Saharan African countries. Is that region unusually prosperous or poor? Is the economy more or less dependent on agriculture or trade? Are there population-wide measures of public health that would help the reader understand how the Kanga province compares to other parts of sub-Saharan Africa?

2. The tables could formatted differently to make them more concise and easier to read. In Tables 1, 2, and 3, binary variables the Ns and %s could be reported for one category since the Ns and %s for the other categories are implied. This would reduce the number of rows. In Table 3, it might not be necessary to provide separate rows and RRs of 1.00 for the reference category.

3. The authors used a step-wise regression approach to choose what variables to include in their final models. For this stage in the research, that might be fine (and the authors do present the information for bi-variate associations), but as the authors move forward in their program of research, I would recommend they move toward making model-specification decisions based more on theory or prior research.

4. I suspect that the way the authors coded age may have resulted in not adequately adjusting for age as a confounder when assessing the unique effect of being out-of-school. Age is a strong correlate of alcohol and tobacco initiation. The fairly wide age bins that were used here may not be ideal. Being out of school is likely to be strongly associated with age. It might be nice to present that information (i.e., the average ages of the in-school and out-of-school groups). There is a concern that the out-of-school group may be made up of adolescents aged 18 or 19; the age 15–19 category might not adequately capture that.

7. PLOS authors have the option to publish the peer review history of their article (what does this mean? ). If published, this will include your full peer review and any attached files.

**Do you want your identity to be public for this peer review?** For information about this choice, including consent withdrawal, please see our Privacy Policy .

Reviewer #2: **Yes: ** Charles Fleming

---

## [Decision Letter · Decision Letter 2]

Determinants of Substance Use Among Adolescents Aged 10-19 Years in Tanga Region, Eastern Tanzania: A Cross-Sectional Analysis.

PGPH-D-24-02968R2

Dear Mr Kessy,

We are pleased to inform you that your manuscript 'Determinants of Substance Use Among Adolescents Aged 10-19 Years in Tanga Region, Eastern Tanzania: A Cross-Sectional Analysis.' has been provisionally accepted for publication in PLOS Global Public Health.

Best regards,

Julia Robinson

Executive Editor

Reviewer Comments (if any, and for reference):

Reviewer's Responses to Questions

**Comments to the Author**

1. If the authors have adequately addressed your comments raised in a previous round of review and you feel that this manuscript is now acceptable for publication, you may indicate that here to bypass the “Comments to the Author” section, enter your conflict of interest statement in the “Confidential to Editor” section, and submit your "Accept" recommendation.

Reviewer #2: All comments have been addressed

2. Does this manuscript meet PLOS Global Public Health’s publication criteria ? Is the manuscript technically sound, and do the data support the conclusions? The manuscript must describe methodologically and ethically rigorous research with conclusions that are appropriately drawn based on the data presented.

Reviewer #2: Yes

3. Has the statistical analysis been performed appropriately and rigorously?

Reviewer #2: Yes

4. Have the authors made all data underlying the findings in their manuscript fully available (please refer to the Data Availability Statement at the start of the manuscript PDF file)?

Reviewer #2: Yes

5. Is the manuscript presented in an intelligible fashion and written in standard English?

Reviewer #2: Yes

6. Review Comments to the Author

Reviewer #2: The authors addressed all of the concerns I raised in my earlier review. The information on the Tanga Region will be helpful for readers. I thought that controlling for age as a continuous variable rather than using fairly wide age bins might change the results, but it did not make much difference. Still, it is better to use the full information on age in the regression models.

7. PLOS authors have the option to publish the peer review history of their article (what does this mean? ). If published, this will include your full peer review and any attached files.

**Do you want your identity to be public for this peer review?** For information about this choice, including consent withdrawal, please see our Privacy Policy .

Reviewer #2: **Yes: ** Charles B. Fleming
